# Modulatory Effects of Atractylodin and β-Eudesmol on Human Cytochrome P450 Enzymes: Potential Drug-Drug Interactions

**DOI:** 10.3390/molecules28073140

**Published:** 2023-03-31

**Authors:** Artitaya Thiengsusuk, Tullayakorn Plengsuriyakarn, Kesara Na-Bangchang

**Affiliations:** 1Graduate Studies, Chulabhorn International College of Medicine, Thammasat University, Pathumthani 12120, Thailand; 2Center of Excellence in Pharmacology and Molecular Biology of Malaria and Cholangiocarcinoma, Thammasat University, Pathumthani 12120, Thailand; 3Drug Discovery and Development Center, Office of Advanced Science and Technology, Thammasat University, Pathumthani 12120, Thailand

**Keywords:** atractylodin, β-eudesmol, metabolic drug interaction

## Abstract

Atractylodin and β-eudesmol, the major bioactive compounds in *Atractylodes lancea*, are promising candidates for anti-cholangiocarcinoma. The inhibitory effects of both compounds on human rCYP1A2, rCYP2C9, rCYP2C19, rCYP2D6 and rCYP3A4 enzymes were investigated using luminogenic CYP450 kits. The modulatory effects were investigated in mouse livers following a daily oral dose of atractylodin or β-eudesmol at 100 mg/kg body weight for 1, 7, 14, and 21 days. The inhibitory effects of both compounds on all rCYP450s were weak (IC_50_: 167 to >686 µM). β-Eudesmol showed the most potent inhibitory effect on rCYP2C19 (IC_50_ = 172.7 µM) and rCYP3A4 (IC_50_ = 218.6 µM). Results of the ex vivo study showed that short exposure (1–7 days) of atractylodin and β-eudesmol resulted in the upregulation of mRNA. Prolonged exposure to the daily oral dose for at least 14 days significantly downregulated the expressions of mRNA and proteins, which correlated with the decrease in the activities of mCYP1A2 and mCYP3A11. Based on the results of the ex vivo study, clinical uses of atractylodin or β-eudesmol for the treatment of cholangiocarcinoma are of concern for the risk of toxicity due to hCYP3A4 inhibition following chronic dosing, as well as the metabolic interaction with the coadministered drugs that are metabolized by hCYP3A4.

## 1. Introduction

Plants and their constituents have been used in traditional medicine to treat various conditions [1]. Globally, using herbal products as complementary and alternative medicine for disease prevention and treatment has been growing [2]. Herbal products are generally considered safe as they have been used for hundreds of years to treat ailments. However, their diverse and complex chemical compositions may cause adverse interactions when used in combination with other herbal or prescription drugs. This is particularly of concern for metabolic drug interactions with cytochrome P450 enzymes (CYP450s), which could result in increased toxicity and/or decreased clinical efficacy of concurrently administered drugs. CYP450s are enzymes essential for the biotransformation of xenobiotics and endobiotics in humans. Approximately 90% of prescription drugs are metabolized by CYP450s [3]. The oxidation of these drugs is most commonly associated with the CYP3A4, CYP2C9, CYP2D6, CYP2C19, and CYP1A2 isozymes [4]. Patients with multiple diseases frequently receive concurrent treatment with several medications. This may modulate the activities of these CYP450s, resulting in clinically significant drug-drug interactions. Investigation of potential CYP450 modulatory effects is therefore crucial.

Atractylodin and β-eudesmol are bioactive sesquiterpenoids isolated from the rhizomes of *Atractylodes lancea* (AL) (Thunb.) DC. [5]. Both are the primary active ingredients responsible for the majority of the pharmacologic properties of AL [6]. Recently, promising activities against cholangiocarcinoma of atractylodin and β-eudesmol have been demonstrated in a series of non-clinical studies [6,7,8,9,10,11,12,13,14,15]. With regard to metabolic drug interactions, the crude ethanolic extract of AL rhizomes was shown to potently inhibit human CYP1A2 in vitro, with comparable potency with the reference hCYP1A2 inhibitor—α-naphthoflavone. In addition, it also moderately inhibited hCYP2C19 and weakly inhibited hCYP2D6 and hCYP3A4 [16]. Human liver microsomes treated with acid-base extracted AL showed a concentration-dependent inhibitory effect on hCYP3A4 enzyme activity [17]. Results of the modulatory effects of atractylodin and β-eudesmol on hCYP450s from various studies remain inconclusive. The current study investigated the inhibitory propensity of both compounds on the five major human CYP450 isoforms—CYP1A2, CYP3A4, CYP2C9, CYP2C19, and CYP2D6 using an in vitro luminogenic CYP450 assay. Their modulating activities on the expressions of mRNA, protein, and enzyme activities of mCYP1A2 and mCYP3A11 were also investigated in mice using real-time PCR, western blot analysis, and enzyme activity assays, respectively.

## 2. Results

### 2.1. In Vitro Inhibitory Effects of Atractylodin and β-Eudesmol on Five Major CYP450 Enzymes

Results of the in vitro inhibitory effects of atractylodin, β-eudesmol and selective inhibitors on five major rCYP450 activities are shown in Table 1. The CYP450 inhibitory potency was classified into three levels based on their IC_50_ (50% inhibitory concentration) values: potent (IC_50_ ≤ 10 µM), moderate (IC_50_ 10–50 µM), and weak (IC_50_ ≥ 50 µM) [18]. The inhibitory effects of atractylodin and β-eudesmol on all rCYP450s were weak compared with the reference compounds (IC_50_: 167 to >686 µM). β-eudesmol showed the most potent inhibitory effect on rCYP2C19 (IC_50_ = 172.7 µM) and rCYP3A4 (IC_50_ = 218.6 µM).

### 2.2. Ex Vixo Modulatory Effects of Atractylodin and β-Eudesmol on the Expression Levels of mCYP1A2 and m3A11 mRNA, Proteins and Enzyme Activities

#### 2.2.1. mRNA Expression

The fold-changes of mRNA expression levels of mCYP1A2 and mCYP3A11 in the livers of mice following daily oral dose of 100 mg/kg body weight of atractylodin or β-eudesmol for 1, 7, 14, and 21 days are shown in Figure 1A,B, and Appendix A.

The expressions of mCYP1A2 and mCYP3A11 mRNA were highest one day after dosing and gradually decreased. For mCYP1A2, atractylodin treatment for 1 and 7 days significantly upregulated mCYP1A2 mRNA expression by 1.6- and 1.2-fold, respectively (Figure 1A). Prolonged treatment for 14 and 21 days, however, significantly downregulated mCYP1A2 mRNA expression by 0.89- and 0.71-fold, respectively. β-eudesmol treatment for 1, 7 and 14 days significantly upregulated mCYP1A2 mRNA expression by 2.63, 1.44, and 1.37-fold, respectively. Prolonged treatment for 21 days, on the other hand, significantly downregulated mCYP1A2 mRNA expression by 0.91-fold. For mCYP3A11, atractylodin treatment for one day significantly upregulated mCYP3A11 expression by 1.72-fold. Prolonged treatment for 14 and 21 days significantly downregulated mCYP3A11 mRNA expression by 0.87- and 0.62-fold, respectively. β-Eudesmol treatment for 1 and 7 days significantly upregulated mCYP3A11 mRNA expression by 2.1- and 1.5-fold, respectively. Prolonged treatment for 21 days, on the other hand, significantly downregulated mCYP3A11 mRNA expression by 0.78-fold.

#### 2.2.2. Protein Expression

Figure 2A–D and Appendix A display the western immunoblot analysis and protein expression ratios of mCYP1A2 and mCYP3A11 in liver microsomes of mice receiving atractylodin or β-eudesmol for 1, 7, 14 and 21 days compared with the control group. mCYP1A2 protein expression levels were significantly downregulated after 21 days of exposure to atractylodin and β-eudesmol (Figure 2A). mCYP3A11 protein expression levels were significantly downregulated after 7, 14, and 21 days of exposure to atractylodin and after 14 and 21 days of exposure to β-eudesmol (Figure 2C).

#### 2.2.3. Enzyme Activities

Phenacetin and nifedipine are substrates of CYP1A2 and CYP3A11, respectively. Both substrates are converted via the CYP enzyme to paracetamol and dehydronefedipine, respectively. The concentrations of paracetamol and dehydronefedipine were used to determine the activities of mCYP1A2 and mCYP3A11. The effects of atractylodin and β-eudesmol on mCYP1A2 and mCYP3A11 activities were, respectively, determined by measuring paracetamol (mCYP1A2-dependent metabolite) and dehydronifiedipine (mCYP3A11-dependent metabolite) concentrations. Figure 3A,B and Appendix A show paracetamol and dehydronifiedipine concentrations in the liver microsomes of mice receiving oral atractylodin or β-eudesmol at a daily dose of 100 mg/kg body weight for 1, 7, 14, and 21 days compared with the control group.

For mCYP1A2, atractylodin exposure significantly increased the enzyme activity one day after administration (Figure 3A). Prolonged exposure for 21 days, however, resulted in a significant decrease in mCYP1A2 activity with lower paracetamol concentrations compared with the control. β-eudesmol significantly decreased mCYP1A2 activity only with prolonged exposure for 21 days. For mCYP3A11, atractylodin exposure for all time periods resulted in a significant reduction in the number of metabolites produced (Figure 3B). β-eudesmol treatment for 14 and 21 days significantly reduced mCYP3A11 activity.

## 3. Discussion

CYP450 constitutes a large protein family that plays a significant role in the metabolism of both xenobiotics and endobiotics, including herbal products and their constituents. Its clinical relevance with therapeutic efficacy and toxicity resulting from the metabolic interactions between the coadministered drugs has garnered a lot of attention. Metabolic interactions between the coadministered medications that are metabolized by the same CYP450 isoforms may occur if they have the characteristics of substrates, inhibitors, or inducers of the same CYP450 isoforms. The most extensively studied CYP450 isoforms are CYP1A2, CYP2C9, CYP2C19, CYP2D6, and CYP3A [19,20]. In the present study, the propensity of atractylodin and β-eudesmol, the main components of *Atractylodes lancea* (AL), to inhibit the activity of major CYP450 enzymes was investigated in vitro using a bioluminescent-based CYP450 assay with human rCYP1A2, rCYP2C9, rCYP2C19, rCYP2D6, and rCYP3A4 enzymes. The P450-Glo assay is one method for detecting dose-dependent rCYP450 inhibition by test compounds. Proluciferins, which are prosubstrates for firefly luciferase and luminogenic CYP450 substrates, were utilized in the assay. This method is rapid and does not require sample preparation before detection by HPLC or mass spectrometry (MS). Furthermore, the method is more sensitive than the majority of fluorescent-based CYP450 assays and eliminates any interference from fluorescent analytes [21].

Among all CYP450s investigated using recombinant CYP450s, atractylodin most potently inhibited rCYP1A2 activity compared to α-napthoflavone, a rCYP1A2 inhibitor, although with moderate potency (IC_50_: 497.2 µM or 90.6 µg/mL). β-Eudesmol on the other hand, most potently inhibited rCYP2C19 (IC_50_: 172.7 µM or 38.4 µg/mL) and rCYP3A4 (IC_50_: 218.6 µM or 48.6 µg/mL) activities. The inhibitory effects of both compounds on rCYP2C9 and rCYP2D6 were very weak (IC_50_: 443 to >685 µM and > 685 µM, respectively). All rCYP450 selective inhibitors ∝-naphthoflavone, sulfaphenazole, troglitazone, quinidine, and ketoconazole, produced the inhibitory effects with the potency of activities comparable to that reported in earlier studies using the same assays [21]. The study was the first to investigate the modulatory effect of atractylodin and β-eudesmol on CYP2C9. In the previous study using human liver microsomes [16], the crude ethanolic extract of AL was shown to potently inhibit hCYP1A2 activity (IC_50_: 0.36 µg/mL). The inhibitory effects on hCYP2C29 and hCYP3A4 were moderate (IC_50_: 16.5 and 54.4 µg/mL, respectively), while the inhibitory effect on hCYP2D6 was weak (IC_50_: 313.5 µg/mL). The observation was partly in agreement with the results of the ligand-binding difference spectroscopy study [22], showing that β-eudesmol bound to hCYP1A2 and hCYP3A4 but not hCYP2C9, hCYP2C19, and hCYP2D6. β -Eudesmol could be the substrate but not the inhibitor of CYP2C19. These results suggest the contribution of other constituents of AL on CYP450 inhibition. The observation of the relatively more potent inhibitory effect of AL on hCYP1A2 could be due to the contribution of other constituents of AL. Altogether, available data suggest that the risk of adverse metabolic drug interactions due to CYP450 inhibition is low when AL as a crude extract or as purified compounds (atractylodin or β-eudesmol) are coadministered with other drugs or herbal medicines. Based on the US FDA recommendations, compounds/extracts with IC_50_ ≤ CYP450 selective inhibitors may have the potential to inhibit CYP450 enzymes clinically [23].

The mouse model was used to investigate the modulatory effects of atractylodin and β-eudesmol on the expressions of CYP450s at three levels, i.e., genes, proteins and enzyme activities. Mice were reported to have nearly all functional genes that are similar or identical to humans, especially with 36 pairs of CYP450 genes [24]. mCYP1A2 in mice is expressed only in the livers, not in extrahepatic tissues [25]. The most similar mouse CYP3A isoform to human CYP3A4 is mCYP3A11, with 76% amino acid homology similarity [26]. mCYP3A11 has been shown to encode the most abundant hepatic mCYP3A protein in male mice [27]. Mouse liver mCYP1A2 and mCYP3A11 were therefore selected for examining the modulatory effects of atractylodin and β-eudesmol as available data indicates that mCYP1A2 and mCYP3A11 enzymes in the mouse livers are clinically implicated in human CYP450 [28,29]. Results of the ex vivo study showed that short exposure (1–7 days) of atractylodin and β-eudesmol resulted in the upregulation of mRNA, while modulatory effects on protein expression and the activities of mCYP1A2 and mCYP3A11 enzymes varied depending on CYP450 isoform. Prolonged exposure to the daily oral dose for at least 14 days, however, significantly downregulated the expressions of mRNA and proteins, which correlated with the decrease in the activities of mCYP1A2 and mCYP3A11. The two compounds were similar in their effects. It was noted that the changes in the expression levels of mRNA of mCYP1A2 and mCYP3A11, in some cases, were not consistent with the proteins and enzyme activities, possibly due to the effects of atractylodin and β-eudesmol on the post-transcriptional or post-translational level.

It is well known that CYP1A2 plays a significant role a key factor in stimulating the metabolism of several carcinogenic chemicals, such as heterocyclics, aflatoxin B1, aromatic amines, and some nitro-aromatic compounds [30]. Pre-carcinogen conversion to carcinogens is more actively activated when CYP1A2 is induced [31]. Some drugs, such as tagreen, that are metabolized by CYP1A2, produce metabolites that can cause hepatotoxicity [32]. Therefore, the substances that induce CYP1A2 may increase the risk of cancer development. On the other hand, those that have the potential to inhibit CYP1A2 can be used as a chemoprotective agent to lessen hepatotoxicity [33,34]. This inhibitory effect is expected to protect against cancer and reduce hepatotoxicity [35]. The CYP3A subfamily is the most prevalent and important group of CYP450s. It involves the metabolism of both xenobiotics and endogenous substances, such as steroid hormones and bile acids [36,37]. CYP3A family contributes significantly (50%) to the bioavailability of therapeutic drugs currently on the market [38]. Based on the results of the ex vivo study, clinical uses of atractylodin or β-eudesmol for the treatment of cholangiocarcinoma is of concern for the risk of toxicity due to CYP3A4 (equivalent to mCYP3A11) inhibition following chronic dosing, as well as the metabolic interaction with the coadministered drugs that are metabolized by CYP3A4.

## 4. Materials and Methods

### 4.1. Chemicals

Atractylodin and β-eudesmol were purchased from Chengdu Biopurify Phytochemicals Ltd. (Sichuan, China) and Wako (Osaka, Japan), respectively. The selective CYP450 inhibitors α-naphthoflavone, quinidine and TBCIP/NBT liquid substrate system were obtained from Sigma Chemicals (St. Louis, MO, USA). Sulfaphenazole and troglitazone were supplied by Cayman Chemical Company (Ann Arbor, MI, USA). Ketoconazole was provided by Tokyo Chemical Industries (Tokyo, Japan). TRIzolTM reagent, SuperScriptTM III Reverse Transcriptase kit and DEPC-treated water were supplied by Invitrogen Life Technologies Inc. (Carlsbad, CA, USA). iTaq Universal SYBR Green Supermix and 2x laemmli sample buffer were purchased from Bio-Rad Laboratories Inc. (Hercules, CA, USA). RQ1 RNase-Free DNase and P450-Glo™ Screening System (Cat. V9770, V9790, V9880, V9890, and V9920) were supplied by Promega (Madison, WI, USA). HPLC grade methanol was supplied by Fisher Scientific (Waltham, MA, USA). Pierce™ BCA Protein Assay Kit was obtained from Thermo Fisher Scientific Inc. (Waltham, MA, USA). Caffeine, phenacetin and paracetamol were supplied by Sigma-Aldrich (St. Louis, MO, USA). Dehydronifedipine was supplied by Toronto Research Chemical (Toronto, ON, Canada). Diazepam was purchased from Government Pharmaceutical Organization (Bangkok, Thailand). Rabbit Anti-mouse CYP1A2 polyclonal antibody and mouse anti-mouse CYP3a11 monoclonal antibody were purchased from ABclonal (Woburn, MA, USA) and Santa Cruz (Dallas, TX, USA), respectively. Rabbit Anti-mouse GAPDH antibody, Goat Anti-Mouse IgG AP-linked and Goat Anti-Rabbit IgG AP-linked were purchased from Cell Signaling Technology (Danvers, MA, USA).

### 4.2. Treatments and Animals

Eight-week-old male ICR (Imprinting Control Region) mice weighing 35–40 g were used to investigate the modulatory effects of atractylodin and β-eudesmol on mCYP1A2 and mCYP3A11. The study protocol was approved by the Ethics Committee for Animal Research at Thammasat University (protocol number 016/2020). All animals were obtained from Nomura Siam International Co., Ltd. (Bangkok, Thailand). The animals were acclimatized for one week before the experiments. A total of 36 mice were allocated to three groups (i) control group (3 mice), (ii) atractylodin-treated group, and (iii) β-eudesmol-treated group. For atractylodin and β-eudesmol-treated groups, animals were fed with 100 mg/kg body weight of each compound for 1, 7, 14, and 21 (3 mice for each dose). Experiments were performed under the guidelines for the care and use of laboratory animals in a controlled environment of 12 h dark/light cycle, temperature 22 ± 2 °C, and relative humidity 30–70% at the Thammasat University Animal Center [39]. Animals were observed daily for changes in signs and symptoms by a veterinarian. Mice were allowed access to the food pellets and water ad libitum.

### 4.3. In Vitro Inhibitory Effects on rCYP450s

The luminescent cytochrome P450 assay was used to determine CYP450-mediated (CYP1A2, 2C9, 2C19, 2D6, and 3A4) metabolism of atractylodin and β-eudesmol. The assay consisted of each recombinant human CYP450 (rCYP450), control membranes lacking P450 activity, a luminogenic CYP450 substrate (luciferin-ME substrate for rCYP1A2, luciferin-H substrate for rCYP2C9, luciferin-H EGE substrate for rCYP2C19, luciferin-ME EGE substrate for rCYP2D6, and luciferin-IPA substrate for rCYP3A4), KPO4 buffer, luciferin free water, NADPH regeneration system, and a luciferin detection reagent. The rCYP450 enzymes were generated from a baculovirus-infected insect cell expression system. The concentrations of the recombinant CYPs used were 0.5 (rCYP1A2 and rCYP2C9), 0.25 (rCYP2C19 and rCYP2D6) or 0.2 (rCYP3A4) pmol (0.5μL) per reaction. In brief, 12.5 µL of rCYP450 reaction mixture (CYP450 enzyme, specific luminogenic CYP450 substrate, and KPO_4_ buffer) was pre-incubated with 12.5 µL of a compound or selective CYP450 inhibitor for 10 min at 37 °C in an opaque white 96-well plate. A luciferin product was produced after adding 25 µL of NADPH Regeneration System (solution A, solution B, and luciferin-free water) and incubated at 37^o^C for 10 (rCYP1A2 and rCYP3A4), 20 (rCYP2C19 and rCYP2C9) or 30 (rCYP2D6) minutes. The reaction was stopped and converted to a luminescent signal after adding Luciferin Detection Reagent (50 µL) and incubated at 25 °C for 20 min. The luminescence signals obtained were determined using a Varioskan^®^ Flash microplate reader (Waltham, MA, USA). The magnitude of luminescence in all of the samples represented CYP450 activity. Stock solutions of all test compounds were dissolved in acetonitrile, except quinidine, which was dissolved in luciferin-free water. All stock solutions were diluted with luciferin-free water until the final concentration of organic solvent was less than 0.25% (*v*/*v*) of the total reaction. The concentration range of atractylodin and β-eudesmol used were 2.74–686 µM and 2.25–562 µM, respectively. Specific CYP450 inhibitors for rCYP1A2, rCYP2C9, rCYP2C19, rCYP2D6, and rCYP3A4 used in the experiments were α-naphthoflavone (0.01–5 µM), sulfaphenazole (0.025–50 µM), troglitazone (0.03–25 µM), quinidine (0.0025–50 M) and ketoconazole (0.0125–12.5 µM), respectively. Total rCYP450 activity was represented by values from untreated wells, and the background luminescence of the assay was represented by values from rCYP450 that lacked activity (minus-P450 control). The concentration that inhibited the activity of each rCYP450 by 50% (IC_50_) was calculated using Calcusyn software (Cambridge, UK). Data are reported as the median (range) of the three independent experiments.

### 4.4. Ex Vixo Modulatory Effects of Atractylodin and β-Eudesmol on the Expression Levels of mCYP1A2 and m3A11 mRNA, Proteins and Enzyme Activities

#### 4.4.1. Quantitation of mRNA by Real-Time PCR

Total mRNA was extracted from mouse livers using Trizol according to the manufacturer’s instructions. The concentration of mRNA was quantified by a nanodrop spectrophotometer. The First-Strand cDNA Synthesis Kit was used to generate cDNA from RNA. The CFX96 Touch Real-Time PCR Detection System was used to determine the expressions of mCYP1A2, mCYP3A11, and β-actin genes using the iTaq Universal SYBR Green Supermix. The forward and reverse primer sequences of each gene are shown in Appendix A. The PCR condition and the delta-delta Ct calculation were according to the previously described article [40]. All primers used in this study were synthesized by Gibthai Co. Ltd. (Bangkok, Thailand).

#### 4.4.2. Preparation of Liver Microsomes and Determination of CYP450 Activities

Mouse liver samples were cut into small pieces, and the liver microsomes were prepared as described previously [40]. The concentrations of proteins in the microsomes were determined using Pierce™ BCA Protein Assay Kit. mCYP1A2 and mCYP3A11 activities were evaluated based on phenacetin O-deethylation and nifedipine oxidation activities, respectively [16]. The incubation mixture of 500 µL consisted of microsomes, substrate (20 mM phenacetin or 40 mM nifedipine), and NADPH. Internal standards (500 µM caffeine or 250 µM diazepam) were added following cold methanol to stop the reaction. The final organic solvent concentration in the incubation mixture was not higher than 1%. Each experiment was repeated three times (duplicated each). Determination of the concentrations of metabolites of mCYP1A2 (paracetamol) and mCYP3A11 (dehydronifedipine) in the reaction was performed using HPLC Agilent 1260 Infinity II with mobile gradient phase between methanol and distilled water at an initial ratio of 20:80 (*v*:*v*) and UV detection (240 nm). Calibration curves for both CYP450 activities were created over a concentration range of 1–50 µM. Oxidized nifedipine concentrations were determined using HPLC as follows: UV detection at 235 nm with an isocratic mobile phase consisting of methanol and distilled water (70:30, *v*:*v*). C-18 reversed-phase column (Thermo Hypersil Gold, 210 × 4.6 mm, 5 µm particle size) with a 1.0 mL/min flow rate was applied for both separations. The linearity of the calibration curve for each analytical assay for CYP450-mediated metabolite production was demonstrated by a determination coefficient (r^2^) of greater than 0.995. In each analytical batch, quality control (QC) samples were run in duplicate at low, medium, and high concentrations. Acceptance criteria were four out of six QC analyses falling within 100 ± 15% of the nominal values.

#### 4.4.3. Western Blot Analysis for Protein Quantification

The microsomal protein concentration for both mCYP1A2 and mCYP1A11 was 25 μg protein per lane. Proteins were denatured by combining equal volumes of liver microsomes with sample buffer (1:19 (*v*:*v*) of 2-mercaptoethanol and 2x Laemmle buffer) and heating at 100 °C for 5 min. Proteins were separated using 12% SDS-PAGE according to their size characteristics before being migrated to nitrocellulose blotting membranes. The membranes were then blocked with a blocking solution (5% skim milk in Tris Buffered Saline (pH 7.5) to prevent nonspecific antibody binding. Primary antibodies (mCYP1A2, mCYP3A11, and GAPDH) were incubated overnight at 4 °C, and the secondary antibody was conjugated to alkaline phosphatase at room temperature for 2 h. The membrane was washed three times for 5 min with washing buffer (TBS pH 7.5 and 0.5% Tween 20), and the bands were generated using a BCIP/NBT solution in the dark without shaking until a band was observed. Protein bands were imaged using UVP ChemStudio (Analytik Jena, Germany), and individual bands were quantified using ImageJ software (National Institutes of Health).

### 4.5. Statistical Analysis

The quantitative data are presented as the median (range or 95% confidence interval) for non-normally distributed data. The Mann–Whitney U test was used to examine the difference between the two quantitative groups. The statistical significance level was set at α = 0.05 for all tests. The analyses were performed using SPSS Version 25 (IBM, NY, USA).

## 5. Conclusions

Clinical uses of atractylodin or β-eudesmol for the treatment of cholangiocarcinoma are of concern for the risk of toxicity due to hCYP3A4 (equivalent to mCYP3A11) inhibition following chronic dosing, as well as the metabolic interaction with the coadministered drugs that are metabolized by hCYP3A4.

## Figures and Tables

**Figure 1 molecules-28-03140-f001:**
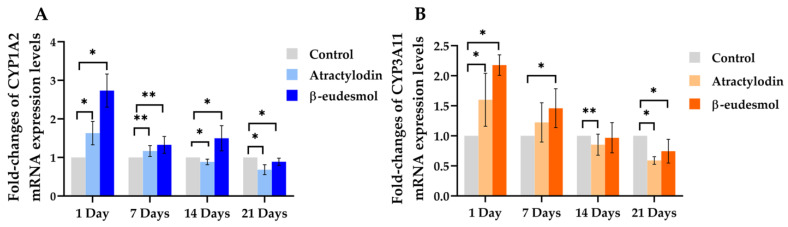
Fold-changes of mCYP1A2 and mCYP3A11 mRNA expression levels in mouse livers. The mRNA expressions of mCYP1A2 (**A**) and mCYP3A11 (**B**) in mouse livers were investigated in male ICR mice following a daily oral dose of 100 mg/kg body weight of atractylodin or β-eudesmol for 1, 7, 14, and 21 days. Control mice were fed with distilled water. Data are expressed as median (95% CI) from three experiments (duplicate each). Asterisks indicate significant differences from the control. Statistical significance with * *p* = 0.002, ** *p* = 0.04 compared with control.

**Figure 2 molecules-28-03140-f002:**
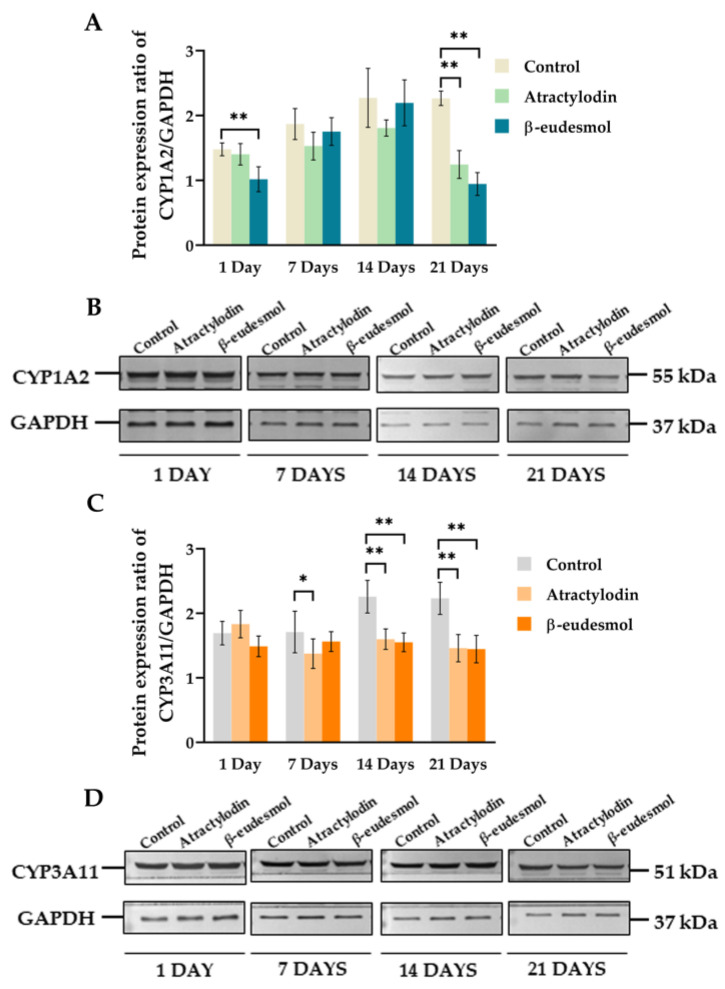
Protein expression ratio and western immunoblots analyses of mCYP1A2 and mCYP3A11 in mouse livers. The protein expression ratio and analysis of western immunoblots of mCYP1A2 (**A**,**B**) and mCYP3A11 (**C**,**D**) were investigated in the polled livers of three male ICR mice following a daily oral dose of 100 mg/kg body weight of atractylodin, or β-eudesmol for 1, 7, 14, and 21 days. Control mice were fed with distilled water. Data are expressed as median (95% CI) from six replications. Asterisks indicate significant differences from the control. Statistical significance with * *p* = 0.039, ** *p* = 0.004 compared with control.

**Figure 3 molecules-28-03140-f003:**
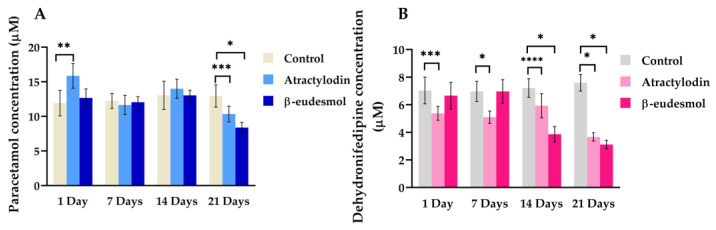
The mCYP1A2-and mCYP3A11-mediated metabolites in mouse liver microsomes. mCYP1A2 (**A**) and mCYP3A11 (**B**) enzyme activities were determined in the liver microsomes of male ICR mice following a daily oral dose of 100 mg/kg body weight of atractylodin or β-eudesmol for 1, 7, 14, and 21 days. Control mice were fed with distilled water. Data are expressed as median (95% CI) from three experiments (duplicate each). Asterisks indicate a significant difference with control. Statistical significance with * *p* = 0.004, ** *p* = 0.006, *** *p* = 0.010, **** *p* = 0.016 compared with control.

**Table 1 molecules-28-03140-t001:** The IC_50_ values of atractylodin, β-eudesmol and the selective inhibitors on the five major human CYP450s (rCYP1A2, rCYP2C9, rCYP2C19, rCYP2D6, and rCYP3A4) enzymes. Data are presented as median (range) values from three independent experiments (triplicate each).

Compounds	Median IC_50_ (Range)				
	rCYP1A2	rCYP2C9	rCYP2C19	rCYP2D6	rCYP3A4
Atractylodin	497.2 (460–561.4) µM	>685.9 µM	>685.9 µM	>685.9 µM	>685.9 µM
	90.6 (83.8–102.3) µg/ml	>125 µg/ml	>125 µg/ml	>125 µg/ml	>125 µg/ml
β-Eudesmol	>562.1 µM	533.3 (443.4–539.2) µM	172.7 (167.3–180.3) µM	>562.1 µM	218.6 (204.6–223.5) µM
	>125 µg/ml	118.6 (98.6–119.9) µg/ml	38.4 (37.2–40.1) µg/ml	>125 µg/ml	48.6 (45.5–49.7) µg/ml
Inhibitors	0.18 (0.13–0.19) µM	0.24 (0.21–0.25) µM	2.07 (1.96–2.33) µM	0.015 (0.013–0.016) µM	0.10 (0.10–0.11) µM
	∝-Naphthoflavone	Sulfaphenazole	Troglitazone	Quinidine	Ketoconazole

## Data Availability

The data presented in this study are available on request from the corresponding author.

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
