# Peer review of "Modulatory Effects of Atractylodin and β-Eudesmol on Human Cytochrome P450 Enzymes: Potential Drug-Drug Interactions"

_molecules, 2023, doi:10.3390/molecules28073140_

Round 1

Reviewer 1 Report

The authors present a study about the modulatory effects of atractylodin and β-Eudesmol on human Cytochrome P450 enzymes. In general, the manuscript will be of interest to those working with cytochrome P450 and metabolic drug interactions. Just some points to deserve attention.

1.- The principal observation is that supplementary material was not included, therefore, it couldnot be revised.

2.- How the 100 mg/kg dose was chosen? Please include this in the text.

3.- Include in methodology a more detailed description about the obtention of the recombinant proteins.

 4.- What was the enzyme concentration or quantity in activity and inhibition assays? In the case of activity assays, it is more important because of the gene and protein expression effects observed.

Author Response

Reviewer 1

  1. The principal observation is that supplementary material was not includes, there, it could not be revised.

Answer:  The key information in the Supplementary Materials are incorporated in the manuscript contents. Detailed information has also been uploaded.

  • “The fold-changes of mRNA expression levels of CYP1A2 and CYP3a11 in the livers of mice following daily oral dose of 100 mg/kg body weight of atractylodin or β-eudesmol for 1, 7, 14, and 21 days are shown in Figures 1A, 1B, and Table S1.”
  • Figures 2A-2D and Table S2 display the western immunoblot analysis and protein expression ratios of CYP1A2 and CYP3a11 in liver microsomes of mice receiving atractylodin or β-eudesmol for 1, 7, 14 and 21 days compared with control group.
  • Figures 3A, 3B and Table S3 show paracetamol and dehydronifiedipine concentrations in the liver microsomes of mice receiving oral atractylodin or β-eudesmol at a daily dose of 100 mg/kg body weight for 1, 7, 14, and 21 days compared with the control group.

  1. How the 100 mg/kg dose was chosen? Please include this in the text.

Answer: The selection of the dose level of 100 mg/kg was based on the maximum tolerated dose of β-eudesmol from our previous acute toxicity. (Reference No. 8: f“Anticancer activity using positron emission tomography-computed tomography and pharmacokinetics of β-eudesmol in human cholangiocarcinoma xenografted nude mouse model”)  For atractylodin, the IC50 values for cytotoxic activity in cancer and normal cells were higher than β-eudesmol (Reference No. 9: “Research and Development of Atractylodes lancea (Thunb) DC. as a Promising Candidate for Cholangiocarcinoma Chemotherapeutics”). Additionally, atractylodin is unstable and is more readily degraded compared with β-eudesmol.  The selection of 100 mg/kg is therefore considered a safe dose to be used in the animal experiments. This is supported by the results showing no adverse effects or toxicity of atractylodin at this dose level throughout the 21-day study period.

  1. Include in methodology a more detailed description about the obtention of the recombinant proteins.

Answer: This information has been added in the Method section as highlighted.

“The rCYP450 enzymes were generated from a baculovirus-infected insect cell expression system.”

  1. What was enzyme concentration or quantity in activity and inhibition assays? In case of activity assay, it is more important because of the gene and protein expression effects observed.

Answer: The concentrations of CYP enzymes used in the inhibition assays were according to the manufacturer’s recommendation as follows.   

  • CYP1A2 used 0.5pmol (0.5μl) per reaction
  • CYP2C9 used 0.5pmol (0.5μl) per reaction
  • CYP2C19 used 0.25pmol (0.25μl) per reaction
  • CYP2D6 used 0.25pmol (0.25μl) per reaction
  • CYP3A4 used 0.1pmol (0.1μl) per reaction

The concentrations of recombinant CYPs used were: (This information have been added in section 4.3: In Vitro Inhibitory Effects on rCYP450s)

  • 5 pmol (0.5μl) per reaction: rCYP1A2 and rCYP2C9
  • 25 pmol (0.5μl) per reaction for rCYP2C19 and rCYP2D6
  • 2 pmol (0.5μl) per reaction for rCYP3A4)

The concentrations of CYP enzymes used in the activity assays were according to the manufacturer’s recommendation as follows:

  • CYP1A2 and 3A11: microsomal protein concentration = 0.4 mg/mL (This has been added in Section 4.4.2: Preparation of Liver Microsomes and Determination of CYP450 Activities)

For Western blot analysis, the microsomal protein concentration used for both CYP1A2 and 3A11 was 25 μg protein per lane. (This has been added in Section 4.4.3: Western Blot Analysis for Protein Quantification)

Reviewer 2 Report

The paper submitted by Artitaya Thiengsusuk and co-workers entitled “Modulatory Effects of Atractylodin and β-Eudesmol on Human Cytochrome P450 Enzymes: Potential Drug-Drug Interactions“ is aimed at evaluation of the effects of the major bioactive compounds in Atractylodes lancea on several human cytochrome P450 isoenzymes. This article represents a well-structured study, which is an important contribution to pharmacological research. The reported data are of sufficient novelty; however, several points should be addressed before publication:

1   1) It is not clear why the authors use lowercase to indicate the subfamily in the case of CYP3a11 isoenzyme, whereas all other enzymes capitals are used. Sometimes, capitals are used in abbreviations of proteins and lowcase – in abbreviations of their genes. The authors use human and mouse enzyme models and different abbreviations for their study, like CYP3A4, rCYP3A4 and CYP3a11. For consistency, I would recommend to introduce organism in every enzyme abbreviation used: hCYP3A4 or mCYP3A11 etc to avoid misunderstandings. The authors should address this issue in the revised version of the manuscript.

2   2) In the “Abstract” and “Conclusions” the authors state that “clinical uses of atractylodin or β-eudesmol for the treatment of cholangiocarcinoma are of concern for the risk of toxicity due to CYP3A4 inhibition following chronic dosing well as the metabolic interaction with the coadministered drugs that are metabolized by CYP3A4”, however, the most of experiments have been done with the most similar to human CYP3A4 mouse CYP3a11. This should provide this information in the abstract and conclusion sections to avoid misleading interpretations.

3  3) It is not clear why the authors give the IC50 values as medians (ranges) rather than standard deviations.

4  4) Lines 120-121. “The effects of atractylodin and β-eudesmol on CYP1A2 and CYP3a11 activities were determined by measuring paracetamol (CYP1A2-dependent metabolite) and dehydronifiedipine (CYP3a11-dependent metabolite) concentrations”. More information should be provided to this experiment. I guess the drugs were phenacetin and nifiedipine. Were the drugs co-administrated?

Author Response

Reviewer 2

  1. It is not clear why the authors use lowercase to indicate the subfamily in the case of CYP3a11 isoenzyme, whereas all other enzymes capitals are used. Sometimes, capitals are used in abbreviations and lowcase- in abbreviations of their genes. The authors use human and mouse enzyme models and different abbreviations for their study, like CYP3A4, rCYP3A4 and CYP3a11.

For consistency, I would recommend to introduce organism in every enzyme abbreviations used hCYP3A4 or mCYP3A11 etc to avoid misunderstanding. The authors should address this issue in the revised version of the manuscript.

Answer: Thank you for this valuable comment. This has been revised as suggested.

  1. In the “Abstract” and “conclusions” the authors state that “clinical uses of atractylodin or β-eudesmol for the treatment of cholangiocarcinoma are of concern for the risk of toxicity due to CYP3A4 inhibition following chronic dosing well as the metabolic interaction with the coadministered drugs that are metabolized by CYP3A4”, however, the most of experiment have been done with the most similar to human CYP3A4 mouse CYP3a11. This should provide this information in the abstract and conclusion sections to avoid misleading interpretations.

Answer:  The Abstract, Discussion, and Conclusion sections have been revised as suggested.

“Based on results of the ex vivo study, clinical uses of atractylodin or β-eudesmol for treatment of cholangiocarcinoma is of concern for the risk of toxicity due to CYP3A4 (equivalent to mCYP3A11) inhibition following chronic dosing, as well as the metabolic interaction with the coadministered drugs that are metabolized by CYP3A4”.

  1. It is not clear why the author give the IC50 values as medians (range) rather than standard deviations.

Answer: Sample size has a significant effect on sample distribution. With the small sample size of 6, the data are not normally distributed. In such case, non-parametric statistical analysis is the appropriate approach and the data are summarized as median (range) instead to mean (SD) which is applied for normal distributed data.

  1. Line 120-121. “The effect of atractylodin and or β-eudesmol on CYP1A2 and CYP3a11 activities were determined by measuring paracetamol (CYP1A2-dependent metabolite) concentration”. More information should be provided to this experiment. I guess the drug were phenacetin and nefiedipine. Were the drug co-administrated?

Answer: The information has been added in Section 2.2.3: Enzyme Activities

“Phenacetin and nifedipine are used as substrates of CYP1A2 and CYP3A11, respectively. Both substrates are converted via the CYP enzyme to paracetamol and dehydronefedipine, respectively. The concentration of paracetamol and dehydronefedipine was used to determine CYP1A2 and CYP3A11, respectively.

  1. Check the Reference lists

Answer: The Reference section has been reordered (Reference 24-41, highlighted in yellow).